Capsules of the diatom Achnanthidium minutissimum arise from fibrillar precursors and foster attachment of bacteria

Leinweber Katrin 1 2 3 katrin.leinweber@uni-konstanz.de
Kroth Peter G. 3
1 Konstanz Research School Chemical Biology , Germany
2 Zukunftskolleg at the University of Konstanz , Germany
3 Biology Department, University of Konstanz , Germany
Smidt Hauke
Electronic publication date: 2015 Mar 26
Publication date: 2015
Volume: 3
Electronic Location ID: e858
Received 2014 Dec 18; Accepted 2015 Mar 5
Copyright: © 2015 Leinweber and Kroth
Copyright year: 2015
Copyright holder: Leinweber and Kroth
License: This is an open access article distributed under the terms of the Creative Commons Attribution License, which permits unrestricted use, distribution, reproduction and adaptation in any medium and for any purpose provided that it is properly attributed. For attribution, the original author(s), title, publication source (PeerJ) and either DOI or URL of the article must be cited.
License URL: https://creativecommons.org/licenses/by/4.0/

Keywords: Biofilms, Diatoms, Scanning electron microscopy, Energy-dispersive x-ray spectroscopy, Diatom-bacteria interactions

Funding: Konstanz Research School Chemical Biology GSC 218 This work was financed by the Konstanz Research School Chemical Biology, the University of Konstanz as well as its Zukunftskolleg. The funders had no role in study design, data collection and analysis, decision to publish, or preparation of the manuscript.

==============================
Achnanthidium minutissimum is a benthic freshwater diatom that forms biofilms on submerged surfaces in aquatic environments. Within these biofilms, A. minutissimum cells produce extracellular structures which facilitate substrate adhesion, such as stalks and capsules. Both consist of extracellular polymeric substance (EPS), but the microstructure and development stages of the capsules are so far unknown, despite a number of hypotheses about their function, including attachment and protection. We coupled scanning electron microscopy (SEM) to bright-field microscopy (BFM) and found that A. minutissimum capsules mostly possess an unstructured surface. However, capsule material that was mechanically stressed by being stretched between or around cells displayed fibrillar substructures. Fibrils were also found on the frustules of non-encapsulated cells, implicating that A. minutissimum capsules may develop from fibrillar precursors. Energy-dispersive X-ray (EDX) spectroscopy revealed that the capsule material do not contain silicon, distinguishing it from the frustule material. We furthermore show that bacteria preferentially attach to capsules, instead of non-encapsulated A. minutissimum cells, which supports the idea that capsules mediate diatom-bacteria interactions.

Introduction

Diatoms (Bacillariophyceae) are among the most productive photoautotrophic, aquatic microorganisms. They contribute an estimated 40–45% to the net primary production (NPP) of the oceans (Mann, 1999), which themselves contribute approx. 45–50% to the global NPP (Field et al., 1998). Additionally, diatoms are important for the biogeochemical cycling of silicon, due to their ornate cell walls. These are called “frustules” and are composed of biomineralised silica (Bradbury, 2004). Cell division includes the separation of the two frustule parts (“thecae”) along a “girdle” region. Each daughter cell then complements its inherited epitheca with a newly synthesised, smaller hypotheca. Within these thecae, slits (called “raphes”) and pores may be present, facilitating the secretion of extracellular polymeric substances (EPS; Wetherbee et al., 1998; Wang et al., 2000). This in turn conveys substrate attachment and motility to benthic diatoms, which often form biofilms with other photoautotrophic algae, as well as heterotrophic bacteria (Buhmann, Kroth & Schleheck, 2012).

The diatom Achnanthidium minutissimum (Kützing) Czarnecki (1994) is a cosmopolitan freshwater diatom (Round & Bukhtiyarova, 1996) that is also found in the littoral zone of Lake Constance. It represents a dominant species complex of early colonisers (Johnson, Tuchman & Peterson, 1997), forming epilithic biofilms in association with a variety of satellite bacteria (Bahulikar, 2006). Such bacteria as well as their spent media have been shown to increase the secretion of extracellular polymeric substances (EPS) like carbohydrates by A. minutissimum (Bruckner et al., 2011). Additionally, growth as well as secretion of proteins and dissolved free amino acids was reportedly influenced in the presence of bacteria (Bruckner et al., 2008; Bruckner et al., 2011). Analogous to the rhizosphere of terrestrial environments (composed largely of fungi and bacteria associated with plant roots), a “phycosphere” has been defined as the space surrounding algal cells including the multitude of inter-kingdom interactions between bacteria and algae (Bell, Lang & Mitchell, 1974; Amin, Parker & Armbrust, 2012).

EPS secretion is of ecological relevance, contributing for instance to the stabilisation of sediments (Cyr & Morton, 2006; Lubarsky et al., 2010). Studying biofilm formation can therefore assist in the understanding of shore and coastline erosion as a result of climate-related changes (see section 3.2.1 of Widdows & Brinsley, 2002 plus references therein). At the same time diatom settlement is one of the major causes of biofouling of man-made machinery in aquatic applications (Molino & Wetherbee, 2008).

A. minutissimum is an excellent model for studying diatom biofilms, because this alga is abundant in natura (Patrick & Reimer, 1966; Krammer & Lange-Bertalot, 1991) and can be cultivated in the laboratory both as “xenic” biofilms (Myklestad et al., 1989) and “axenic” suspension cultures (Windler et al., 2015). Xenic cultures contain bacteria from the diatom’s natural habitat. Removal of these bacteria is possible (for example by antibiotic treatment) and yields viable axenic cultures (Bruckner & Kroth, 2009; Windler, Gruber & Kroth, 2012). Bacteria-free cultures allow the establishment of bioassays in order to study the interactions between diatoms and bacteria, although potentially unwelcome long-term effects have to be taken into account. For example, axenic cultivation can lead to a reduction of average cell size and to frustule deformations (MacDonald, 1869; Pfitzer, 1871; Geitler, 1932; Windler et al., 2014). However, such deformations also occur naturally and can be used as stress markers, as well as water quality indicators (Morin, Coste & Hamilton, 2008; Falasco et al., 2009; Cantonati et al., 2014).

A. minutissimum forms biofilms and extracellular structures like stalks and capsules. These structures have been defined as “unidirectionally deposited, multi-layered structures attaching cells to substrata” (Hoagland et al., 1993). Stalks have been investigated previously by transmission electron microscopy and biochemical techniques to elucidate their structural morphology and chemical composition (Daniel, Chamberlain & Jones, 1987). Additionally, a phase model of diatom adhesion involving stalks has been developed (Wang et al., 1997): Stalks may protrude from so called “basal pads” of aggregated EPS at the apical valve faces within hours to a few days, thus elevating the cells above the substrate. Capsule formation in A. minutissimum biofilms occurs later in the stationary phase, is possibly triggered by bacterial influences, and may cement diatom attachment (Windler et al., 2015). That study also found that axenic A. minutissimum cultures mostly secrete soluble carbohydrates while the insoluble carbohydrates in xenic cultures were positively correlated with the appearance of capsules.

Diatom capsules have puzzled phycologists for a long time, and their potential physiological and ecological function have elicited a variety of hypotheses (Lewin, 1955; Geitler, 1977). For example, capsules have been proposed to participate in locomotion, flotation, attachment, waste removal, catchment of inorganic nutrients, storage of polysaccharides, sexual reproduction, as well as protection against grazing and dehydration. More recently, it was demonstrated that capsule formation is dependent on at least “a certain minimum light intensity,” sparking the idea that capsules might serve as an additional polysaccharide storage pool, once intracellular capacities are saturated (Staats et al., 2000).

Diatom capsules and other extracellular material have been characterised biochemically, while electron microscopy has enabled highly detailed morphological characterisations of diatom frustules (Toyoda et al., 2005; Toyoda et al., 2006; Morin, Coste & Hamilton, 2008). Additionally, atomic force microscopy (AFM) revealed many mechanical properties of the extracellular polymers of some diatoms (Crawford et al., 2001; Higgins et al., 2003; Willis et al., 2013). In the present study, scanning electron microscopy (SEM), as well as energy-dispersive X-ray (EDX) spectroscopy were employed to analyse the microstructure and development stages of Achnanthidium minutissimum capsules in order to further develop this species as a model system for diatom-bacteria interactions, and to elucidate one aspect of the complex interactions of diatoms and other microorganisms.

Materials and Methods

Cultivation conditions

Achnanthidium minutissimum (Kützing) Czarnecki (1994) was isolated from photoautotrophic epilithic biofilms of Lake Constance as previously described (Windler, Gruber & Kroth, 2012). Stock cultures were grown in cell culture flasks with ventilation caps (Sarstedt, Newton, North Carolina, USA) filled with modified liquid Bacillariophycean Medium (BM; Schlösser, 1994; Windler, Gruber & Kroth, 2012) in two different culture states: either with co-isolated bacteria (“xenic”), or “axenic” after their removal (Windler, Gruber & Kroth, 2012). Monthly, these stock cultures were scraped off the flask bases and sub-cultured in new BM.

Biofilms were grown directly on SEM sample carriers by the following procedure: Sample carrier disks of ca. 1 cm in diameter were punched from Thermanox tissue culture cover slips (Miles Laboratories Inc., Elkhart, Indiana, USA). Thermanox material has two different sides, therefore care was taken to always store and handle the disks right-side-up. They were sterilised by immersion in 70% isopropanol (v/v in H2O) over night and subsequently irradiated with UV light for 2 h in a laminar flow cabinet. One sterile disk was placed into each well of 6-well plates (Sarstedt, Newton, North Carolina, USA, order number 83.1839.500) and covered with 3 to 5 mL BM. Culture wells were inoculated with 5 × 105 to 1 × 106 A. minutissimum cells from the stock cultures after those were checked to be axenic or xenic. Axenicity was confirmed by SYBR Green staining and observation under a BX51 (Olympus, USA) bright-field fluorescence microscope using GFP fluorescence filters. Well plates were sealed with Parafilm and incubated at 16 °C under an illumination regime of 12 h dark and 12 h light at 20–50 µmol photons × m−2 × s−1 for 11 to 31 days.

Crystal violet staining and bright-field microscopy

Thermanox disks were removed from stationary cultures after 11 to 31 days with inverted (“soldering” or “cover glass”) forceps (Hammacher, Germany) and rinsed with 1 mL sterile-filtered tap water. A Gram-staining protocol adapted from Kaplan & Fine (2002) was applied to visualise adherent cells and their extracellular polymeric structures as follows: A droplet of 200 µL solution of 0.02% crystal violet (CV) in sterile filtered tap water was applied onto the disk for 12 min, which was held suspended by forceps. Disks were rinsed with 1 to 3 mL water, until the runoff no longer contained visible CV. In order to find the same cell clusters in both microscopic approaches, pointing or encircling scratches were made into disk surfaces.

Disks were placed on moistened glass slides and moistened additionally with 20 µL sterile-filtered tap water. Cover slips were applied carefully and marked regions were observed under a BX51 (Olympus, Tokyo, Japan) bright-field fluorescence microscope using chlorophyll fluorescence filters. Images of these areas at various magnifications were taken with AxioCams MRm (fluorescence and grey-scale images) and MRc (colour) using AxioVision software (Zeiss, Oberkochen, Germany).

Scanning electron microscopy (SEM) and energy-dispersive X-ray (EDX) spectroscopy

Diatom cells were fixed on Thermanox disks by incubation in a mixture of 2% glutaraldehyde, 10 mM CaCl2 and 10 mM MgCl2 in 0.1 M sodium cacodylate buffer at pH 7 and room temperature (RT) for 2 h. Dehydration was conducted first with 30% and 50% EtOH, at RT for 2 h each, followed by 70% EtOH at 4 °C over night, 90% EtOH at RT for 2 h and finally with 96% and 100% EtOH twice for 1 h each. Critical point drying in CO2 followed (Balzers CPD030; Oerlikon Balzers, Balzers, Liechtenstein) and samples were finally sputtered with gold (Au) and palladium (Pd) to a thickness of 5 nm (Balzers SCD030; Oerlikon Balzers, Balzers, Liechtenstein).

After fixation, dehydration and Au/Pd-sputtering, the biofilm-covered Thermanox disks were imaged with a Zeiss “AURIGA” scanning electron microscope, controlled with the “SmartSEM” software v05.04.05.00. The elemental composition of samples was analysed by energy-dispersive X-ray (EDX) spectroscopy. Samples were excited with the AURIGA’s electron beam at 10 kV and the emitted X-rays (of specific energy levels due to the elemental electron configuration) were recorded with an Oxford Instruments “X-Max 20 mm2” detector (Oxford Instruments, Scotts Valley, California, USA) and the “INCA” software v4.15.

Bacteria counting and data visualisation

Bacteria (rod-shaped particles) on fully visible A. minutissimum valve faces were counted in scanning electron micrographs. Valve faces were classified into frustules and capsules, depending on whether pores were visible or completely disappeared under a layer of capsule material. Diatom cells with partial encapsulation were not included in the counting, and neither were bacteria cells which attached to the girdle bands of diatom cells.

ImageJ v1.46r with the Cell Counter plug-in v2010/12/07 was used to count diatoms and bacteria cells. This data was evaluated and visualised with R v3.1.1 (language and environment for statistical computing; R Development Core Team, 2011), ggplot2 v1.0.0 (Wickham, 2009) and RStudio v0.98 Desktop Open Source Edition.

Results and Discussion

For BFM and SEM observations, we cultured Achnanthidium minutissimum xenically and axenically on Thermanox disks. After incubation periods of 11 to 31 days, the disk surfaces in xenic cultures were densely covered by a mono-layer of A. minutissimum cells (Fig. 1). This biofilm was visible by eye as a light greenish-brown coloration on the substrate disks after removal from the medium. Staining with the dye crystal violet (CV) and subsequent bright-field microscopy showed that large portions of the diatom cells were surrounded by capsules.

Figure 1 Crystal violet (CV) stained capsules (grey ovals) in xenic A. minutissimum biofilm (scale bar: 20 µm).

Micrograph depicts 11 days old culture and is a merge of the chlorophyll fluorescence channel (red; indicating diatom cells) and the bright-field image (grey). Some mature capsules are marked with arrows. Bright spots within diatom cells are lipid bodies. Bacteria are visible as light and dark speckles around and in between the diatom cells.

In contrast, axenic A. minutissimum cells did not form biofilms, so that even careful rinsing left much fewer cells attached to the disks and thus available for SEM analysis. This observation is in agreement with studies that utilised other growth substrates to compare biofilm formation by axenic and xenic diatom cultures. By measuring chlorophyll contents, the possibility that axenic cells might simply be less proliferate was excluded (Windler et al., 2015). Xenic A. minutissimum cultures on the other hand have also been found to develop biofilms on glass beads as well as in plastic multi-well plates (Lubarsky et al., 2010; Windler et al., 2015). Our results demonstrate, that xenic biofilms of A. minutissimum can also be grown on Thermanox disks, enabling direct preparation for electron microscopy of native biofilm samples.

Identification of A. minutissimum capsule microstructures

In order to correlate the hydrated A. minutissimum capsules visible in light microscopy to their dehydrated appearance in SEM, areas were marked by scratches on the CV stained disks and cells of interest were identified by BFM. Subsequently, the same areas and cells were found again in SEM (Fig. 2). The same technique was successfully applied to axenic cultures, despite the lower prevalence of adherent cells (Fig. S1).

Figure 2 Identification of A. minutissimum capsules (asterisks) by successive observation of cell clusters by first bright-field and then scanning electron microscopy of xenic biofilm (scale bars: 5 µm).

(A) Bright-field micrograph of crystal violet (CV) stained, 31 days old culture. Encapsulated cells (asterisks) are strongly stained, while weak staining indicates few extracellular polymeric substances (EPS) on the frustule surfaces. (B) Scanning electron micrograph of the the same cell cluster. Encapsulated cells (asterisks) are surrounded by an opaque material. Frustule pores are visible on cells that did not possess a capsule in the hydrated biofilm. Note also the unequal distribution of bacteria cells on capsules versus non-encapsulated frustules.

In BFM, the CV stained capsules were visible as voluminous, rounded structures around most of the cells. As extracellular polymeric structures in the genus Achnanthidium are composed mostly of carbohydrates (Wustman et al., 1998; Windler et al., 2015), strong hydration in the native biofilm is likely the source of this appearance of the capsules. In SEM, we were able to distinguish two types of A. minutissimum cells in xenic biofilms already at low magnifications: cells with pores in their frustules still visible, and cells covered by an apparently unstructured material masking the pores.

The frustules of non-encapsulated xenic, as well as axenic A. minutissimum cells appeared identical to those from scanning electron micrographs shown in previous studies (Mayama & Kobayasi, 1989; Potapova & Hamilton, 2007; Hlúbiková, Ector & Hoffmann, 2011). The low prevalence of raphes in our images is most likely due to their orientation towards the substrate for mucilage secretion (Gordon & Drum, 1970; Wetherbee et al., 1998). Natural attachment and orientation of cells on our biofilm disks was retained because we did not employ harsh preparation techniques, such as boiling the diatom cells in sulphuric acid (Mayama & Kobayasi, 1989). Such harsh treatments are designed to prepare only frustules and in our case would likely have resulted in cell detachment from the growth substrate, as well as random orientation on the SEM sample carrier. Instead, we utilised the SEM sample carrier disks directly as growth substrates for the biofilms.

The SEM images in Fig. 3 show that the capsule material appears to be unstructured or slightly granular, resembling the “adhering film and tube” of Cymbella microcephala and Cymbella prostrata reported in figures 31 and 32 of Hoagland et al. (1993) and also the shaft ultra-structure of the marine diatom Achnanthes longipes displayed in Fig. 8 of Wang et al. (2000).

Figure 3 Comparison of microstructures on A. minutissimum cell surfaces in a xenic biofilm.

(A) Capsule material is sometimes stretched between cells and/or towards the substrate (arrows; scale bar: 2 µm). Culture was 11 days old at the time of fixation for SEM. Asterisks denote magnified areas B and C. (B) Non-encapsulated cells possess a fibrillar mesh of varying degrees of density. Frustule pores are only partially covered and in some cases, fibrils stick out from the frustule (scale bar: 1 µm). (C) Encapsulated cells are completely covered with a material of slightly granular structure, but lacking clearly discernible features (scale bar: 1 µm).

In addition to covering the cells, the capsule material also had sheet-like structures (arrows in Fig. 3A) where it was stretched between A. minutissimum cells and the anchoring points on the substrate. This pattern is most likely due to dehydration during SEM sample preparation (Hoagland et al., 1993). Due to fixation of the samples prior to drying, the hydrated capsules most likely shrank in their entirety.

Fibrillar precursors may give rise to A. minutissimum capsules

Closer inspection provided many examples that the frustules of non-encapsulated xenic A. minutissimum cells were not completely free of extracellular polymeric substances (EPS). Instead, they were covered by a mesh of fibrils (Fig. 3B), arranged mostly around the frustule pores, sometimes crossing them and sometimes sticking out. The average diameter of these fibrils was about 45 ± 9 nm. The fibrils were rarely observed to be secreted through the pores, although these were found to be large enough (from 60 to 140 nm in diameter), showing round to elongated shapes. Fibrils were generally longer than the diameter of pores, but quantification was not performed because branching and interweaving of the fibrils made it impossible to determine the respective beginnings or ends. To the best of our knowledge, this is the first report of frustule-attached fibril structures in freshwater diatoms. Similarly structured, thinner fibrils were reported previously only for marine diatoms (Bosak et al., 2012).

Fibrillar meshes of varying densities were detected in both axenic and xenic cultures, but only the latter also contained capsules. Axenic cultures appeared to contain more cells with few, short fibrils (Fig. S2), which may initiate surface attachment of cells in both culture types. However, only in xenic cultures, bacteria could induce the secretion of firmer EPS structures (Bruckner et al., 2011), and thus substrate adhesion, in the majority of cells.

Furthermore, we found intermediate stages between the fibrillar meshes that covered the frustule surface only partially and the complete encapsulation with apparently unstructured material (Fig. 4). The similarity in surface structure suggests a relation of fibrils and capsules. The disordered arrangement of fibrils shown in Figs. 4A and 4B is also a feature of the unstructured capsule material shown in Fig. 4C. In it, no particular order of the slightly granular substructures is discernible either. This visual impression suggests that the fibrillar meshes on A. minutissimum cells might be precursors to capsules, into which the fibrils may condense. An alternative explanation for the capsule structure may be the polymerisation of a secondary type of fibrils upon the primary mesh (Fig. 4A), relegating the latter to a scaffolding function.

Figure 4 Scanning electron micrographs of terminal parts of A. minutissimum cells at potentially different encapsulation stages within xenic biofilms (scale bars: 1 µm).

Fibrillar meshes (A) may form capsule material (C) by denser growth and cross-linking of fibrils (B). Depicted samples were taken from stationary, 11 to 31 days old cultures.

In either case, the fibril formation is not strictly synchronised between cells in the same culture. It rather appears to be a function of each cell’s individual age, because different stages appeared both on days 11 and 31 of incubation, as well as in between.

To further elucidate whether or not fibrils and capsules might be related, we analysed mechanically stressed capsule areas (Fig. 5). Here, tension yielded an alignment of capsule microstructures, as well as fraying on the edges. Fibrillar structures resulting from these processes were similar in diameter to the frustule-covering fibrils.

Figure 5 Fibrillar microstructures (arrow pairs) within capsule material of A. minutissimum cells in xenic biofilm are revealed by mechanical stress (scale bars = 1 µm).

Micrographs depict samples from an 11 days old culture. (A) Tip of a partially encapsulated cell. Fibrillar substructures are continuous throughout the capsule material. (B) Capsule material stretched between cells frays into fibrils.

There are two possible sources for the mechanical force. Firstly, motility of the cells relative to each other in the native biofilm. However, it has been reported for a related species in the order Achnanthales, that motility is lost upon production of EPS structures (Wang et al., 1997). Secondly, mechanical force could be caused by the dehydration during SEM sample preparation. Both explanations lead to the question why mechanical force highlighted the fibrillar microstructure, while relaxed capsule areas (see previous figures) appeared unstructured. Micromanipulation experiments may be required to further investigate the properties of A. minutissimum capsules and their fibrillar microstructures. Such experiments have been conducted by force-mode AFM on the mucilage layers of other diatoms (Higgins et al., 2003; Willis et al., 2013).

Based on this data, we suggest fibrils as a precursor candidate for capsules. Fibrils may condense into the capsule material as depicted in Fig. 4, for example by enzymatic cross-linking or transglycosylase activity. Fibrils may be disguised in relaxed capsule material because they are arranged in a disorderly fashion, but mechanical stress yielded a visible alignment.

Capsule material does not contain silicon

Silicon (Si) is a major component of diatom frustules, in which it is present as hydrated silicon dioxide. Capsules on the other hand may consist mostly of extracellular polymeric carbohydrates (Wustman et al., 1998; Windler et al., 2015). In order to exclude the possibility that the capsule material we observed might represent frustule deformations or extensions (Windler et al., 2014; Cantonati et al., 2014) it was screened for the presence of Si. Energy-dispersive X-ray (EDX) spectra were recorded from capsule areas with and without a cell body and thus frustule below them (Fig. 6).

Figure 6 Capsule material in xenic A. minutissimum biofilms does not contain silicon and can thus be distinguished from the frustule.

Samples were taken from 11 days old culture. (A) Scanning electron micrograph of capsule material that is stretched between A. minutissimum cells (partially visible). Labels indicate energy-dispersive X-ray (EDX) measurement points. (B) EDX spectra show that less silicon (peak around 1.75 keV, with grey background) is found in the capsule material without cell body below it (black trace) compared to the control points on the cell bodies (blue and turquoise traces). Stronger Au signal around 2.15 keV likely results from larger gold-sputtered surface area in close proximity to the “capsule” measurement point.

As expected, higher counts around 1.75 keV, which signifies Si (Guerra et al., 2013; Chandrasekaran et al., 2014), were obtained from capsule material, than from cell bodies. Background level Si counts likely result from the frustule edges closeby due to the “pear effect” (Arnould & Hild, 2007). It explains how excited electrons diffuse into the sample, so that a pear-shaped volume of ca. 0.5–1 µm diameter below the measurement point or area also emits detectable X-rays. The cell bodies in this figure are separated by approximately that distance. The stronger gold (Au) signal of the capsule material around 2.15 keV compared to the control areas probably resulted from the larger sputtered surface area within the measurement volume. Due to the absence of notable Si signals from A. minutissimum capsule material, we can exclude the possibility that it is some kind of frustule extension or deformation.

Furthermore, we can tentatively exclude chitin as a major component of the capsule material, because no notable nitrogen signals (N; 0.39 keV) were recorded from it. Chitin fibrils have been found to be secreted by diatoms into the surrounding water body (Gardner & Blackwell, 1971; Herth, 1979). In contrast, the fibrillar meshes we describe here, tightly covered the frustule surfaces of individual A. minutissimum cells and therefore likely represent different EPS structures.

Bacteria preferentially attach to encapsulated diatom cells

It became apparent during the SEM observations, that diatom-attached bacteria cells occurred more often on capsules than on frustules. To substantiate this observation, bacteria cells were counted on both diatom cell surface types (Fig. 7).

Figure 7 Distribution of the number of bacteria cells adherent to diatom valve faces of different surface types (frustule or capsule) in xenic A. minutissimum biofilms.

Bacteria were counted in SEM images, if they were in direct, visible contact with the valve face of either a frustule (N = 54) or a completely encapsulated diatom cell (N = 71; see Figs. 2B and 3A for illustration). Samples were taken from 11 to 31 days old cultures. Boxes represent 1st and 3rd quartile. Black center lines represent medians. Diamond symbols represent means. Whiskers extend to 1.5-fold of the inter-quartile range (IQR). Black dots represent extreme values that lie outside the IQR.

Notably higher numbers of bacteria (ca. 25 times more on average) adhered to capsules than to frustules throughout the stationary phase (means: 11.41 ± 8.23 and 0.46 ± 0.82 respectively). The variance in the numbers of bacteria per diatom was larger (ca. 100 times) on encapsulated cells than on frustules, indicating that not all encapsulated A. minutissimum cells were equally strongly colonised by bacteria. Presumably, this is due to the population dynamics of the co-isolated bacteria within different areas of the xenic biofilms.

Bacteria as well as diatom cells with only fibrillar meshes on their frustule were found to individually retain attachment to the substrate. We deem it unlikely that attachment to each other could have been too weak to withstand the rinsing step. Instead, the bacteria preferentially adhered to capsules, while either being actively repelled from non-encapsulated A. minutissimum cells or only not especially drawn to their frustules.

It has been proposed that A. minutissimum capsules might be an asset in the mutualistic relationship of this diatom with its satellite bacteria (Windler et al., 2015). Previous findings suggest a pattern of bacterial adherence to A. minutissimum cells in xenic biofilms that would support this hypothesis (Windler, Gruber & Kroth, 2012): diatom cells were surrounded by a bacteria-free space, followed by a layer of densely aggregated bacteria cells. Although no CV stains were conducted in that study, the bacteria-free regions resemble the EPS structures reported since then as capsules.

In bacterial biofilms, nutrient distribution is predominantly determined by diffusion, sometimes along strong gradients within a biofilm (Stewart, 2003). Similarly, variations of cellular nutrient distributions within freshwater diatom biofilms exist (Murdock et al., 2010). Furthermore, it is possible that diatom capsules serve as a common nutrient pool to the satellite bacteria in a mutualistic relationship (Bruckner et al., 2008). Therefore, competition between individual diatom cells for re-mineralising bacteria could occur. Nutrient-limited, but still photosynthetically active diatom cells may produce predominantly insoluble carbohydrates to foster close attachment of heterotrophic bacteria that re-mineralise EPS or secrete vitamins.

Our finding that bacteria attach preferentially to capsules strengthens the argument that capsules play a role in the inter-kingdom relationship of satellite bacteria and benthic diatoms. Whether this relationship is antagonistic, mutualistic or commensal in nature remains to be elucidated. Labelling experiments with isotopes or fluorophores may assist in the determination of carbohydrate fluxes from the diatom’s EPS structures to bacteria feeding on those.

Supplemental Information

Figure S1 Identification of A. minutissimum cell clusters in axenic culture by subsequent observation by both bright-field (A) and scanning electron (B) micrography (scale bars: 5 µm)

Demonstration of the same technique used to identify the appearance of xenic biofilms and dehydrated capsule material in SEM (main Fig. 2) in axenic cultures after 31 days of incubation with much fewer adherent cells and no capsules.

Click here for additional data file.

Figure S2 Scanning electron micrographs of fibril-covered A. minutissimum frustules from axenic culture

Samples were prepared for SEM after 20 days of incubation. (A) (scale bar: 1 µm) & (B) (scale bar: 200 nm): Frustules with few, short fibrils, which were not found in xenic biofilms. C (scale bar: 1 µm) & (D) (scale bar: 200 nm): Frustule with medium-dense fibrillar mesh, as also seen in xenic biofilm (main Fig. 4A). (E) (scale bar: 200 nm): Fibrils are not only flatly attached to the frustule but also stick out into space and make contact with other cells (arrows), as also seen in xenic cultures (main Fig. 4B).

Click here for additional data file.

We thank Joachim Hentschel, Lauretta Nejedli and Michael Laumann of the Electron Microscopy Center of the University of Konstanz for sample preparation, SEM and EDX device operations, and insightful discussions, as well as Ansgar Gruber and Carolina Rio Bartulos for helpful ideas and suggestions. Our gratitude also belongs to two anonymous reviewers whose valuable suggestions improved this manuscript greatly.

Additional Information and Declarations

Competing Interests

Author Contributions

Data Deposition

The authors declare there are no competing interests.

Katrin Leinweber conceived and designed the experiments, performed the experiments, analyzed the data, contributed reagents/materials/analysis tools, wrote the paper, prepared figures and/or tables, reviewed drafts of the paper.

Peter G. Kroth contributed reagents/materials/analysis tools, discussed the design of the project, and reviewed drafts of the paper.

The following information was supplied regarding the deposition of related data:

Raw data and R script by which Fig. 7 was generated are available at http://dx.doi.org/10.6084/m9.figshare.1273931.

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
