# Peer review of "Capsules of the diatom Achnanthidium minutissimum arise from fibrillar precursors and foster attachment of bacteria"

_PeerJ, doi:10.7717/peerj.858_

## Round 0.1 · original submission · Major Revisions

As you will see below, both reviewers agree that your paper is very well written, and that experimental procedures are sound. Having said that, both provide a number of more general and specific suggestions for further improvement, which I would ask you to address in a revised manuscript. Please also specifically address the unavailability of reference Windler et al., 2014b, which seems to be a thesis report submitted to the University of Konstanz. My suggestion would be to incorporate the necessary information in the manuscript, as it will otherwise be difficult for the reader to repeat your experiments.

Reviewer 1 ·

Basic reporting

The studies of diatom extracellular polysaccharides, especially their role in the diatom-bacteria interactions, are a very timely and interesting subject and this article represents a very valuable contribution to this exciting field of research. The article is generally well written and easy to follow, and I am impressed by the thoroughness of the study and the detailed report. The introduction is clear and comprehensive, and the results are very well described and discussed appropriately although there is a need to cut some superfluous parts. Therefore, I would suggest several minor modifications of the manuscript prior to acceptance.
The figure descriptions need some improvement and my specific comments and suggestions are listed below under Comments to the author. I would suggest removing Figure 6 and the accompanying section of the results as these observations are not necessary, and the image quality is only mediocre in comparison to other figures.
The literature should be revised carefully and uniformed according to the journal criteria. In the list of references some have the article title with all capital letters, others do not. Also, please be careful to italicize the species names in the titles. Several references are unclear and missing information, unfortunately among them are the ones most relevant to the manuscript (e.g. Windler et al. 2014b).

Experimental design

Methods are scientifically sound and precisely described. However, I would suggest several improvements and modifications as in some parts more details are necessary while in others some information is redundant (see comments below).

Validity of the findings

The conclusions are appropriately stated and supported by data, although some parts could be cut and the length of the manuscript slightly reduced.
The presentation of LM and SEM images side by side is really nice and impressive evidence on the same structures visualized by the different methodology.
I have only a remark on the part where the SEM experiment was described with frustules subjected to the higher excitation voltage. This whole section is redundant as well the Figure 6 and should be removed. The question was if the material described as capsule was indeed an organic structure, and this is well supported with the data of EDX spectra which do not show any of Si in the structure. The excessive voltage in SEM would burn everything, make organic structures as well as silica or carbonate material melt and translucent if you apply it long enough. I do not see the point in presenting such an experiment here as you have more than enough evidence to support your hypothesis.

Additional comments

I am glad for the opportunity to review this interesting work and I hope my remarks will be helpful. Please find below my detailed comments on each point:
Abstract
Please insert in the first sentence the information of the freshwater origin of A. minutissimum. Rephrase the sentence near the end where you state that the capsule does not arise from the cell wall, you showed that it is not the same as the cell wall, but organic material of which capsule is formed is extruded from the cell and closely adjoined with the frustule, so you can say only that it is possible to distinguish the capsule and the frustule material and not speculate of the capsule origin.
Introduction
L 25 – the former name is not relevant here as the is not a taxonomic paper and it is enough to cite the valid authorities: Achnanthidium minutissimum (Kützing) Czarnecki 1994 (the year of publication is 1994 as it appears in Algaebase, but please check the validity)
L 29 – the mentioned studies by Bruckner 2008 and 2011 investigated specifically the species A. minutissimum interactions with bacteria or other diatoms? If so, please specify.
L33 –L 44 – are there any studies where the diversity of the bacterial species and their taxonomic affiliation was investigated in xenic cutures or natural habitat (bacteria interacting with the A. minutissimum)?
L 50 – “like”, you mean stalks?, sentence unclear
L 71-72 - there are many EM investigations on the diatom frustules as EM is the irreplaceable tool in diatom studies (Round et al. 1990) so I would suggest to list here the studies which focus on the frustule morphology of A. minutissimum.
Material & methods
L81 – remove the previous name of the species, also list the full name with the authorities only in the introduction when the species is mentioned for the first time
L 81-83 - please include the details on the isolation procedure and cultivation conditions as the reference Windler 2014b is not complete in the reference section and if I understood properly this reference is actually a chapter in a book and this kind of publications are harder to access than the open access journal
L 84 – use abbreviation SEM (you defined it already in the introduction) and delete: - instead 48 well, it is important what you used, not what you did not use.
L 88 – rephrase sentence, describe only how you sterilized the disks and remove “because autoclavation proved to melt them”
L 92 - 93 – rephrase sentence, remove the explanation that you could put several discs in one well and describe only what you have done, merge it with the description above
L 94 – cultures not cells are axenic or xenic, and also you concluded this by the staining procedure and observing the wells with an eye or using a microscope?
L 99 – observations of the cells after 20 days are not reported in the results
L 110 – the light microscope BX1 is operational with both brightfield and fluorescence at the same time, so this is brightfield (remove the dash) fluorescence microscope using chl fluorescence filters . And also, for my personal curiosity, do both Zeiss cameras that you mentioned work well on the Olympus microscopes? And which camera was used for the brightfield and which for the fluorescence image as you mentioned 2 different types of camera, MRm and MRc, are they mounted on the same microscope?
L 112 – remove the statement: “See figure 1 and 2”. It is more clear and appropriate to present the information on the type of microscope used in the description of the figures. The same applies below, lines: 125, 127, 136
Results & Discussion
L 137 – choose either & or and apply it also to the subtitle line 79
L 138 – 141 - integrate this information in the material and method section, its redundant iinfo on the beginning of the results
L 143- The biofilm was visible as light greenish by eye or?
L 146 – rephrase; capsules probably did not absorb chl a fluorescence, but they simply do not fluorescent because they do not contain chlorophyll and there is no excitation of this material, same as the storage material (which is maybe the white dots visible within the cells).
L 154 – 155 – So your results demonstrate that only xenic cultures are possible to grow on discs and observe with SEM?
L 160 – the same areas and cells
L 161 – remove the expression” balloon like structures”, not suitable as balloon only has an outer “skin” and rest of it contains air and the capsule is mucous and the cells are completely embedded in it; you explained it well in the following sentence. Change it where mentioned in rest of the manuscript (e.g. figure descriptions).
L 164 – just a notion, although you applied critical point drying , which should preserve the integrity of the organic material, nevertheless the structure shrinked and became amorphous
L 192 – granularity in appearance? the surface was granular or?
L 197 – 198 - The sentence is unclear
L 209 – so the capsules were not found in the axenic cultures, but fibrillar meshes? Do you have an image of axenic cultures? from LM or from SEM?
L 234 – 238 – remove this paragraph, this is too much speculation and you do not have any evidence
L 244 – add some references in which micromanipulation experiments were done, as you suggested, or remove this part
L 246 – 252 - this is a good and well done experiment, but I think your evidence of the composition of the capsule area is more than enough. Yes, in SEM if you apply very high voltage on a single spot you will burn the silica (or melt it) also the organics.. so I do not see necessary to present this evidence or the Figure 6.
L 277 – 283 – a little bit hard to follow as the mentioned studies were not easily accessible, I would suggest to rewrite this part an first point put the results from the current study and then compare it to the others.
L 292 – 296 yes, possible, also bacteria produce EPS themselves
L 297 – 307 – I find this paragraph presenting too much speculation with no support of the results from the presented study, so I would suggest to remove it.
References
In addition to my previous comments of capital letters and italicized species names :
Bahulikar 2006 – journal is missing, Geittler 1932 please write out the complete title of the journal, Pfitzer 1871 missing publisher and the place published, Windler 2014a is missing the volume and pages, and Windler 2014b is unclear, is it a thesis or a chapter in which book?
Figures:
Fig 1: Mark the capsule with an arrow, and remove “balloon like” description, also the capsule is not absorbing the chl fluorescence , but its not excitated by the Uv lamp of the microscope therfpre it does not fluorescence. What are the bright white dots within the cells?.
Fig 2: I must say that this image is very nice. Be specific only in the description, the asterisks are not marking the capsules, they mark the encapsulated cells, rephrase. What are those two small objects on the non-encapsulated cell in the lower right corner?
Fig 3: In the description: early stationary phase, xenic culture . Is it 11 days?
Fig 4: Improve the description: Detailed view on the terminal part of the cell. Also, could you slightly darken or increase the contrast, the fibrils then should be more visible
Fig 5: Mechanical stress caused by drying? stretching? The description should be self sufficient to explain the figure. Also, mark with an arrow the fibrillar structure in A
Fig 6- remove
Fig 7: Its an image of partial view, not of complete cells. Add the info that the spectrum is measured at the marked spots. Also if you know which cell si turquise and which blue mark it in the image. In the image of the spectrum there is a repetition, in the right corner cell, cell, capsule which is again repeated lower part.
Fig 8: Delete the last sentence, it is redundant. Can you mark in the plot the bacterial counts in frustules and capsules differently (triangles and squares for instance), there are some dots that I cannot attribute where the bacteria were counted.

Reviewer 2 ·

Basic reporting

This manuscript is generally well-written and structured. A major problem with respect to basi reporting however, in my opinion, lies with the fact that in many parts (introduction, materials & methods, results and discussion) reference is made to Windler et al 2014b, and this reference is not accessible (no full reference is given?). This is problematic as it is not possible to check many aspects of the paper (materials & methods, but also some specific results to which reference is made, see below) and also, importantly, it is not possible to check how and to what degree the Windler paper exactly differs from what is presented in the present manuscript (i.e. to check how ‘self-contained’ the present manuscript is).
Some terms should be better explained: for example, for the term phycosphere reference could be made to the Amin et al MMBR 2012 review paper on diatom-bacteria interactions. Also, the ‘capsule’ should be properly defined in the introduction (for this term reference is made to Windler et al 2014b…).
Most figures are clear and necessary. However, it is not clear to me what the variation along the second (Y) axis represents in Fig. 8? Also, I suggest using arrows to show the bacteria in Fig. 1, the continuous fibrillary substructures in Fig. 4A, and the fibrillary microstructures in Fig. 4B.

Experimental design

The treatments and analyses all appear to be sound, but, as mentioned above, for some methods [diatom cultivation, axenisation (actually no method or reference is provided for this at all, except for a reference to Windler et al. 2014b in the introduction on line 43), reference is made to an inaccessible paper (Windler et al. 2014b). Likewise, reference is made to relevant results that can only be found in this paper (lines 57, 150-151, 153, 255, 278, 306). Some results are mentioned on line 197-198 but I don’t see the data/evidence.

Validity of the findings

The results of the manuscript are very interesting but in many cases the evidence presented is rather circumstantial, which makes the conclusions sometimes rather speculative. For example, in line 212-213 it is suggested that axenic cells with very little EPS were probably washed off prior to preparation. It would be easy to verify this by collecting the washed off cells. In the next lines, it is suggested that the bacteria induce EPS secretion. However, higher EPS production might also be caused by nutrient limitation in the presence of competitors (bacteria)? It may be possible to check this by adding spent medium of the bacteria. Line 234-235: what is this statement based on? Line 239-240: this is highly speculative. Would it not be possible to monitor fibril production and EPS secretion in time, after adding the bacteria? Fig. 7: what explains the high P value of the capsular material? Line 292-293: this statement rather contradicts the statement on line 212-213. Line 290-291: the variability in bacterial cell numbers on encapsulated cells is huge: any idea what could cause this? So in general, I think the manuscript shows potentially interesting data but the evidence supporting the claims of the paper (e.g. in the title and abstract).

Additional comments

See major comments above. Minor comments include:
• Line 27: do you mean that A. minutissimum represents a species complex?
• Line 34: I don’t understand what you mean with the biogeographical relevance of EPS secretion.
• Line 50: word missing after ‘like’
• Line 64: insert ‘to’ after proposed
• Line 71: insert e.g. before references
• Lines 70-74. I am sure there are more SEM studies on stalk formation, EPS secretion etc. (see papers by Wetherbee, Pickett-Heaps, etc.)?
• Briefly explain principle of EDX spectroscopy in materials and methods
• Line 131: insert ‘a’ after under
• Line 138-141: not necessary, repeats the materials and methods.
• Line 182: omit ‘be’
• Line 187: were instead of where
• Line 188: impossible instead of infeasible?
• Line 240: omit one ‘by’

---

## Round 0.2 · accepted · Accept

I agree with both reviewers that the revised manuscript addresses all previously raised issues, and that it is now fit for publication.

Reviewer 1 ·

Basic reporting

No comments

Experimental design

No comments

Validity of the findings

No comments

Additional comments

The authors have answered on all of my comments appropriately and I am satisfied with all the changes made. The manuscript has been significantly improved.

Reviewer 2 ·

Basic reporting

No comments.

Experimental design

No comments.

Validity of the findings

No comments.

Additional comments

I have previously reviewed a first version of this manuscript. I went through the rebuttal letter, the revised text and figures, and I am now happy with the new version. I have no further comments, apart from the fact that this is a nice piece of work!